# Examine frameworks policies and strategies for effective information governance in healthcare organizations

Richard Okyere Boadu[1]*, Victor Wireko Adu[1], Kwame Adu Okyere Boadu[2], Baishatu Ibrahim[1], Prince Akey[1], Amponsah Amishadas Mensah[1], Godwin Adzakpah[1], Nathan Kumasenu Mensah[1]

**1** Department of Health Information Management School of Allied Health Sciences, College of Health and Allied Health Sciences, University of Cape Coast, Cape Coast, Ghana, **2** Sunyani Teaching Hospital, Sunyani, Ghana

* richard.boadu@ucc.edu.gh

## Abstract

### Background study

Health information, over the years, has been regarded as one of the crucial assets in the landscape of health on the grounds of the myriads of benefits derived from it. It plays a major role in policy planning and implementation, quality improvement, clinical decision-making, care continuity, disease surveillance, etc., making it an integral part of healthcare delivery. This substantiates the pressing need for an appropriate information governance (IG) system, ensuring proper data stewardship, measures to ensure data quality and that preserving the privacy and confidentiality of patient data are in place. As part of conscious efforts to improve health information governance, this study examines the entire information governance structure of healthcare facilities used as study areas, taking into account the effectiveness of data stewardship, data quality management and compliance with regulatory requirements.

### Methodology

Employing a cross-sectional study design, data was collected from 432 healthcare professionals emanating from Cape Coast Teaching Hospital (CCTH), Eastern Regional Hospital Koforidua (ERHK) and Ledzokuku-Krowor Municipal Assembly (LEKMA) Hospital, all in Ghana. A semi-structured questionnaire, demarcated into five (5) parts, was used to collect the data. Closed-ended questions were analyzed with Stata 15.0 involving frequencies, percentages, means, standard deviation, standard errors and confident intervals. A chi-square test of independence was deployed to ascertain relationships between variables. Open-ended questions were coded and analyzed with Qualitative Data Analysis Miner 6.0, deploying inductive thematic analysis.

**Data availability statement:** All relevant data are within the paper and its Supporting Information files.

**Funding:** The author(s) received no specific funding for this work.

**Competing interests:** The authors have declared that no competing interests exist.

## Results

The level of understanding of data stewardship (2.87/4.0), familiarity with concepts of data quality (2.73/5.0) and ethical considerations and privacy (3.13/5.0) when dealing with patient health data were found to be varied among the professionals, overall rated as moderate and affirming the need for improvement. The level of knowledge and understanding on these pertinent areas were highly influenced by type of profession (p < 0.001; p = 0.031 and, p < 0.001) respectively and been indicative that some type of health workers is much concerned about data stewardship and health information governance than others. Various methods and efforts were in place to ensure the quality of data; however, hampering these attempts included factors such as poor data collection methods, lack of resources, inadequate competency and some behavioral factors. In addition to these factors is the low level of training on data stewardship (1.86/4.0), which heavily leads to skewness in knowledge. The majority of healthcare professionals were familiar with and advocated for adherence to regulatory requirements such as HIPAA (33.7%), GDPR (21.3%) and HITECH Act (14.4%), in making data governance much robust. Efforts were also identified in ensuring privacy and confidentiality during the storage, processing and transmission of patient data mostly by deploying methods such as authentication (58.9%), use of secured computers and software (71.2%) and using secure electronic platforms (66.0%) respectively indicating a keen attention for data stewardship and governance. About 70% of healthcare professionals attested to the regular conduct of monthly compliance audits, however, the rate of training on data stewardship was discovered to be very low (1.86/4.0).

## Conclusion

There is an existence of clear policies and procedures to guide data stewardship to enhance overall information governance, however, healthcare professionals' level of knowledge on subjects of information governance ought to be enhanced. There is a need to intensify training sessions on data quality, and regulatory requirements for health information governance to mitigate the gaps in knowledge among professionals.

## 1. Introduction

### 1.1 Background study

Every nation's health system is designed and developed by its demands and resources. The community should always have access to high-quality services from a desirable health system. All health systems must provide individuals with high-quality care, maintain community health, lower per-capita healthcare costs, and adopt the best policies and decisions based on reliable information [1] Delivering healthcare services is the primary goal of healthcare organizations, and achieving this goal

requires using health information [2]. Healthcare organisations should prioritise the management and control of health information. This goal can be accomplished by utilizing information as a strategic resource in plans and choices made at various levels of the health system [2]. According to [1], to effectively manage information and support organizational strategy as well as operational, legal, safety, and environmental needs, there is a need for information governance (IG), a complete organizational instrument utilized in the health system. In actuality, it is a strategic framework comprising the norms, procedures, roles, and standards that individuals and organizations use to create, arrange, secure, preserve, use, and discard information based on organizational goals. Information Governance focuses on developing an infrastructure made up of rules and guidelines for bettering information management in healthcare organizations [3]. American Health Information Management Association (AHIMA) defines Information Governance as a broad organizational framework for managing information during the life cycle of information, supporting the strategic, operational, legal, and statutory programs and risks in the organization [3]. According to the Association of Records Managers and Administrators (ARMA), IG is a strategic framework including standards, processes, roles, and criteria that hold organizations and individuals accountable for creating, organizing, protecting, using, and disposition of information by the objectives of the organization [4]. Health Information Governance in healthcare organizations is a critical framework that establishes centralized policies, procedures, and accountabilities for managing patient information effectively [5].

In recent decades, due to the increasing adoption of communication and information technologies in healthcare systems, many developed countries and some other countries use health information networks, which has led to growths in data generation and distribution of healthcare information [6]. The existence of various healthcare data generation centers and the requirement of communication and sharing health data between organizations have made it inevitable for related organizations to apply IG for health information management. [6]. Again, the high volume of healthcare data and the need to integrate them strongly necessitate applying the IG approach [7]. According to [8] and [9], developed countries have found out the significance of health IG; for example, in the United Kingdom, the Health and Social Care Information Centre (HSCIC) is responsible for health IG through a self-assessment process. In the healthcare sector, Information Governance focuses on data privacy, confidentiality, and the secure management of patient records and operational information. It encompasses elements like data stewardship, quality management, and compliance with regulatory requirements to ensure trusted information essential for patient engagement and treatment [10]. Establishing a comprehensive Information Governance strategy involves creating cross-functional committees, developing organization-wide strategies, enhancing existing frameworks, empowering employees through training, and striving for continuous improvement through performance metrics [11].

The implementation of effective Information Governance practices remains a pressing challenge for organizations striving to manage and safeguard patient data securely. Despite the critical importance of Information Governance in ensuring data integrity, privacy, and compliance with regulatory requirements, many healthcare organizations still struggle to establish comprehensive frameworks that address the complexities of modern healthcare data management [12]. The lack of immature IG programs in healthcare, as highlighted by recent surveys, poses significant risks to patient confidentiality, data accuracy, and organizational compliance [6]. Without robust Information Governance structures in place, healthcare organizations face heightened vulnerabilities to data breaches, regulatory penalties, and compromised patient care outcomes. [13]. The absence of standardized policies, inadequate data stewardship practices, and insufficient quality management protocols contribute to the fragmentation and inconsistency of information across healthcare systems. The urgency to address these Information Governance deficiencies is underscored by the increasing regulatory scrutiny in healthcare, such as the stringent requirements of HIPAA, which demand stringent data protection measures and accountability from organizations (Hussain Seh et al., 2020). Failure to establish robust Information Governance frameworks not only jeopardizes patient trust but also impedes the ability of healthcare providers to harness the full potential of data-driven insights for delivering high-quality care [14]. In light of these assertions, this study provides insight into the current state of information governance in healthcare, exploring the effectiveness of data stewardship, best practices for data

management, compliance with regulatory requirements, and ethical considerations and potentially proposes recommendations for improving information governance. In addition to this, the study probed into healthcare professionals' level of awareness, understanding and familiarity with these pertinent of subject areas of information governance. Significantly, the study contributes to the ever-growing scholarly literature on how to effectively implement health information governance by unearthing the strengths and weaknesses of already established information governance structures.

### 1.2 Study Hypotheses

To ascertain the influence of professional type and years of experience on the level of awareness, understanding and familiarity, the following hypotheses were investigated:

**H1**: The level of awareness and understanding about data stewardship is dependent on the professional type.

**H2**: The level of awareness and understanding about data stewardship is dependent on years of experience.

**H3**: The level of familiarity with concepts of data quality is dependent on professional type.

**H4**: The level of familiarity with concepts of data quality is dependent on years of experience.

**H5**: The level of familiarity with ethical consideration and privacy concerns is dependent on the professional type

**H6**: The level of familiarity with ethical consideration and privacy concerns is dependent on years of experience

## 2. Study methodology

### 2.1 Study design

A cross-section design was employed, enabling the collection of data from healthcare professionals who generate, use and manage health information in the healthcare facilities under study at a point in time from 15 July 2024 to 15 August 2024.

### 2.2 Study area

The study took place in three different health facilities, all located in Ghana. The Eastern Regional Hospital Koforidua (ERHK) is situated in the Eastern Region of Ghana. ERHK was established in 1926, and it is a secondary-level referral facility for the entire Eastern Region and doubles as a municipal hospital for the New Juaben Municipal with 180,000 inhabitants. The hospital uses the Lightwave health information management system to provide healthcare services to its clients. The hospital has a bed capacity of 356 and offers the following services: Obstetrics and Gynecology; Internal medicine including anti-retroviral therapy, pediatrics, surgery, dental, ophthalmology, physiotherapy, ear, nose and throat, laundry, mortuary, and primary healthcare services. ERHK is a Ghana Health Service Facility, which is a not-for-profit healthcare organization. Ledzokuku-Krowor Municipal Assembly (LEKMA) Hospital was built by the People's Republic of China's Government as a China-Ghana Friendly Hospital. It is a 100-bed capacity hospital that has all the units of a general hospital, including specialist services, laboratory, and radiological facilities. In addition, it has a Malaria Research Centre and Herbal Medicine Unit. The hospital's clinical staff is made up of a team of about 45 native doctors of which 13 are specialists as well as 40 Chinese specialists in the various fields of clinical management, namely, Anaesthesiology, Cardiology, Radiography, Neurology, Local Chinese Medicine, Surgery, Paediatrics, and Obstetrics and Gynaecology among others. Over 100 trained nurses, pharmacists, and paramedical staff, about 60 midwives, 30 laboratory technicians, 35 orderlies, 50 security staff, and more. There are other staff members, namely, the Information technology department, imaging studies department, administrative offices, and many others. Cape Coast Teaching Hospital (CCTH) in the Cape Coast Metropolis of the Central Region of Ghana. The Cape Coast Teaching Hospital is one of the agencies under the Ministry of Health. With a current bed capacity of 400, the hospital is mandated to provide tertiary clinical services, serve as training for graduate and postgraduate medical programs, and undertake research into emerging health problems. It also serves as the referral facility for the health facilities in the Central, Western, and Western North regions of Ghana. It was established in August 1998 as the Central Regional Hospital and later upgraded to a Teaching Hospital status in March 2014, following

the establishment of the School of Medical Science at the University of Cape Coast, Ghana. Cape Coast Teaching Hospital is also accredited for postgraduate training by the Ghana College of Physicians and Surgeons.

## 2.2  Study population

The study involved all healthcare professionals in the facilities under study who generate and use health information for various purposes. The category of professionals includes Doctors, Pharmacists, Health Information Officers, Medical Records Officers, Lab Technicians, Nurses, Administrators, IT Staff, and any other health personnel who create, generate, or use health information for various purposes.

## 2.3  Sampling technique/ procedure

A sample size of 412 was calculated from an estimated population size of 3,312 using Epi Info statistical software version 7.2.5.0 StatCalc function with a Confidence Level = 97%, Expected Frequency = 50%, Acceptable Margin of Error = 5%, Design Effect = 1.0, Clusters = 1. From a stratification based on the category of profession, a simple random sampling technique was deployed in drawing the sample from the targeted population of healthcare professionals in the various health facilities. A proportion of HCP were selected from every stratum through the randomization of staff ID, giving equal opportunity to participate in the study.

## 2.4  Data collection tools and techniques

A semi-structured questionnaire was provided to the intended respondents as a tool to supplement the information requested from the participants. Closed-ended questions represented a quantitative part of the study, while opened-ended questions were also intended for the qualitative aspect. Questionnaires were disseminated in two versions, hardcopy and softcopy, for the respondents' convenience. The questionnaire had five (5) sections with sub-questions. The sections were (1). demographics, (2) effective data stewardship for health information governance, (3) strategies and practices for managing data quality, (4). compliances and regulatory requirements related to health information governance, (5) ethical considerations, and privacy concerns associated with patient data handling. Specifically, open-ended questions solicited qualitative data on current methods used for assessing and measuring data quality, data quality challenges, and the strategies and best practices deployed to address the challenges (all under section/objective 3 of the study). In addition, qualitative data were garnered under section 5, where HCP professionals gave measures/strategies employed by their organization during the storage, transmission, and processing of patient data. Ample time was given to the respondents, at least one day, so that they could carefully complete the questionnaires.

## 2.5  Data analysis

Data was captured, transformed and analyzed using statistical software. Close-ended questions were analyzed using STATA 15. Exploratory analysis included frequencies and percentages, descriptive statistics such as means, standard deviations, standard errors and confident intervals. Further statistical tests were conducted to ascertain the existence of association/dependency between variables using the Chi-Square test of independence. All statistical tests were deemed significant at a 5% alpha level. Open-ended questions were coded and analyzed with Qualitative Data Analysis Miner 6.0, deploying inductive thematic analysis. The analysis also involved some charts, visually presenting some results.

## 2.6  Ethical consideration

An ethical clearance letter with reference number [CCTHERC/EC/ 2024/118] was obtained from the Cape Coast Teaching Hospital Ethical Committee (CCTHEC). Also, permission was sought from the authorities of all the healthcare facilities involved in the study aided by the obtained ethical clearance letter and specific introductory letters. At each interview point, the purpose of the study was comprehensively explained to the participants. A verbal consent was obtained from

the respondents, and they were allowed to decide whether to partake in the study. Those who consented to participate were included in the study. The respondents were assured of the confidentiality of their identity in the study. The verbal consent, which was part of the preamble of the questionnaire and approved by the IRB, was read to the respondents to decide whether to participate in the study.

## 3.1 Results

### 3.1.1 Demographic characteristics of participants

The demographic characteristics of the healthcare professionals involved in our study comprised gender distribution, age category, profession type and years of experience as indicated in Table 1 below. Of a total of 432 healthcare professionals, males were more than half the total, representing about 57.6%. The majority of the professionals were less than 40 years old, representing approximately 85%, and the 15% were 40 years or more.

The top three professional types which dominated the survey were Nurses, 86(19.9%); Health Information Officers, 73(16.9%) and Medical Records Officers, 65(15.1%) of the total number of professionals recruited during the survey.

**Table 1. Demographic characteristics of study participants (n = 432).**

| Variable | Frequency | Percent |
|---|---|---|
| Gender | | |
| Male | 249 | 57.64 |
| Female | 183 | 42.36 |
| Age Category | | |
| 20–29 | 166 | 38.97 |
| 30–39 | 198 | 46.48 |
| 40–49 | 46 | 10.8 |
| 50–59 | 16 | 3.76 |
| Mean = 32.6, min = 20, max = 57, std. dev = 6.9 | | |
| Profession | | |
| Medical doctor | 35 | 8.1 |
| Physician Assistant | 35 | 8.1 |
| Nurse | 86 | 19.91 |
| Midwife | 35 | 8.1 |
| Medical records officer | 65 | 15.05 |
| Health information officer | 73 | 16.9 |
| IT officer | 39 | 9.03 |
| Lab technician | 41 | 9.49 |
| Radiologist and sonographer | 19 | 4.4 |
| No response | 4 | 0.93 |
| Years of experience | | |
| 1–5 | 283 | 65.66 |
| 6–10 | 112 | 25.99 |
| 11–15 | 23 | 5.34 |
| 16–20 | 9 | 2.09 |
| 20+ | 4 | 0.93 |
| Mean = 5.2, Min = 1, Max = 25, Std. Dev = 3.9 | | |

**Source:** *Survey of healthcare professionals, 2024.*

Approximately 65.7% of the professionals had a working experience of 1–5 years, and about 26% had 6–10 years. Only 8.3% of them had more than 10 years.

### 3.1.2 How effective data stewardship for health information governance is established and implemented

The understanding of data stewardship is essential for effective health information governance in the healthcare facility. The results of the study indicated that the majority of the HCP have a moderate understanding of data stewardship, with only 15.5% having a high level of understanding. However, about 78% of the healthcare professionals confirmed that there is an existence of clear policies and procedures to guide data stewardship, but not all the professionals had a clear understanding of the said document. Moreover, as indicated in Table 2, on a scale of 1–5, the average rate of organizing training on data stewardship for healthcare professionals was as low as 1.86. Possibly, the low rate of training for healthcare professionals on data stewardship is responsible for the low average level of awareness and understanding of data stewardship (2.87/4.00) as indicated in Table 2.

A further analysis of how the professional type and years of experience of healthcare professionals influence their level of awareness and understanding on the subject of data stewardship was conducted using the Chi-Square test of independence and effect size measured with Cramer's V. At a 5% alpha level, the test was significant for type of profession ($\chi2 = 104.773$, $p < 0.001$ and Cramer's V = 0.246) as indicated on Table 3. In other words, there is a moderate relationship between profession type and data stewardship, indicating that some healthcare professionals are more concerned and understanding of data stewardship than others.

### 3.1.3 Strategies and best practices for managing data quality for information governance initiatives

Regular conduction of data quality assessment is one of the best practices for managing and governing health information. The response from healthcare professionals during the survey indicated that data quality assessment was mostly

Table 2. Factors indicating the effectiveness of data stewardship in healthcare (n = 432).

| Variable | Frequency | Percent |
|---|---|---|
| Current level of awareness and understanding of data stewardship | | |
| Very low | 51 | 11.81 |
| Low | 40 | 9.26 |
| Moderate | 265 | 61.34 |
| High | 67 | 15.51 |
| No response | 9 | 2.08 |
| *Scale = 1–4, Avg(mean)= 2.87 [2.78–2.95], Std. Dev = 0.887* | | |
| Availability of clear policies and procedures to guide data stewardship | | |
| Yes | 337 | 78.01 |
| No | 15 | 3.47 |
| Not sure | 80 | 18.52 |
| Frequency for organizing training on data stewardship for HCP | | |
| Never | 129 | 29.86 |
| Rarely | 241 | 55.79 |
| Occasionally | 57 | 13.19 |
| Regularly | 5 | 1.16 |
| *Scale = 1–4, Avg(mean)= 1.86 [1.79–1.92], Std. Dev = 0.676* | | |

**Source:** *Author's Analysis of Survey of Healthcare Professionals, 2024.*

**Table 3. Influence of professional type and years of experience on the level of understanding and awareness of data stewardship among healthcare professionals.**

| Variable | Descriptive Statistics | | |
|---|---|---|---|
| | Scale | Mean | Std. Error | 95% CI |
| Current level of awareness and understanding of data stewardship | 1–4 | 2.87 | 0.043 | [2.78 - 2.95] |
| Demographic characteristics | Test of Independence | | |
| | Test Statistic (χ2) | df | p-value | Cramer's V |
| Profession type | 104.773 | 9 | 0.001** | 0.246 |
| Years of experience | 13.631 | 4 | 0.626 | 0.089 |

Source: *Author's Analysis of Survey of Healthcare Professionals, 2024 ** Significant at p<0.001*

conducted on monthly intervals, 286(66.2%). However, 19(4.4%), 26(6.02%) and 17(3.94%) also attested that data quality assessment was conducted on a weekly, quarterly and annual basis, respectively.

The professionals' familiarity with the concept of data quality which is imperative for effective information governance was an average of 2.73[2.62–2.84] on a scale of 1–4 which implies a vast room for improvement as indicated in Table 4. Years of experience are not a significant influencer of the level of familiarity with the concept of data quality in the study, hence, failing to accept hypothesis H4.

In assessing and measuring the quality of data in the various healthcare facilities, the current method that is mostly deployed for this purpose, according to the healthcare professionals, was the conduction of data quality audits and assessments in various units, 34.8%. The organization of monthly data validation meetings was also a common methodology used for assessing the quality of data, as attested by the HCP (30.1%). Moreover, the use of Computer-aided data quality checks was also found to be a method used for measuring and assessing the quality of data, attested by 22.0% of HCP. With the computer-automated data quality checking, validation rules have already been fed into the backend of the computer software, allowing data to be validated when entered. Peer reviews, supervision and monitoring were also methods used by sometimes to assess the quality of data generated in the healthcare facilities, however, they were rarely used as attested by only 13.1% of healthcare professionals. Some of the extracts of the responses are provided below:

"…*we conduct data quality audits and assessment on a regular basis to check data quality*"- Stated by a Doctor

"*The facility organizes data quality assessment and audits as means of assessing data quality. It is mostly led by the health information officers*" – Stated by a Medical Laboratory Technician.

**Table 4. Healthcare professionals' level of familiarity with the concept of data quality and its significance in information governance initiatives in the healthcare settings and its dependence on profession type and years of experience.**

| Variable | Descriptive Statistics | | |
|---|---|---|---|
| | Scale | Mean | Std. Error | 95% CI |
| Familiarity with the concept of data quality and its significance in information governance | 1–5 | 2.73 | 0.559 | [2.62 – 2.84] |
| Demographic characteristics | Test of Independence | | |
| | Test Statistic (χ2) | df | p-value | Cramer's V |
| Profession type | 18.364 | 9 | 0.031* | 0.206 |
| Years of experience | 4.833 | 4 | 0.305 | 0.106 |

Source: *Author's Analysis of Survey of Healthcare Professionals, 2024 * Significant at p<0.05*

*"The District Health Information Management System (DHIMS) that we use has data quality checks installed that help monitor the quality of data captured"- Stated by a Health Information Officer*

*"The electronic health records we use in the facility has in-built data quality checks that ensures that valid data is entered into a field". – Stated by a Midwife*

*"we do manual data quality checks and audits"- Stated by a Medical Records Officer.*

*"…data validation meetings are organized every month for auditing and validating data generated in the various units", – Stated by a Midwife.*

*"Data quality validation which is organized by the hospital on every month to reflect on data inconsistency before final submissions are done by the health information unit.' – Stated by a Doctor*

*"In our facility, we do peer reviews, regular supervisions and monitoring spearheaded by the health information officers to ensure quality of data collection and management" – Stated by a Health Information Officer.*

The above quoted statements are some prominent responses stated by the study participants, indicating the kind of methods used in their respective healthcare facilities for assessing the quality of data generated from various point of care. These quotes reflect the caliber of qualitative data summarized and presented in Fig 1. Healthcare professionals acknowledged the combinations of these methods being used for assessing data quality.

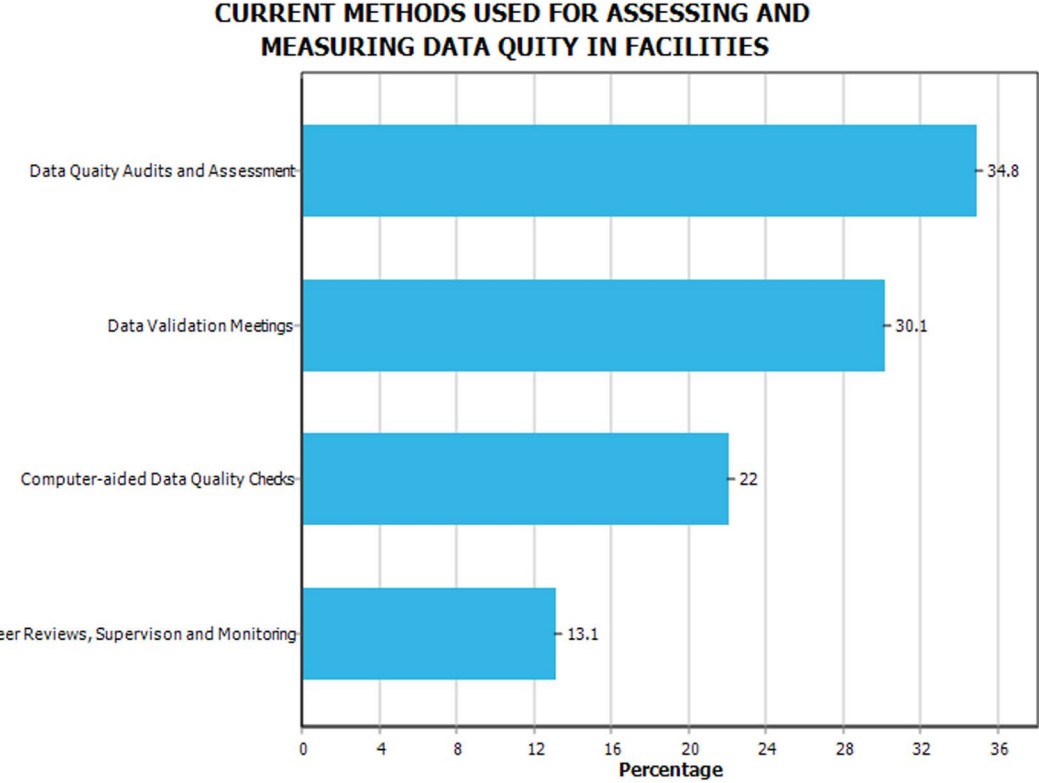

**Fig 1. Current methods used in facilities for assessing and measuring the quality of data for information purposes.**

Various challenges were identified to be hampering efforts towards ensuring quality data in the healthcare facilities. These impediments were categorized under four (4) main themes as displayed on table 5: (1)the use of Poor data collection methods (dominated by "data capturing errors and inconsistency", 80(22.4%); (2)Inadequate resources (with "less number of staffs", 87(24.4%); "limited computer systems, 34(9.5%)"; "Poor Internet connectivity, 16(4.5%)" been the top three sub-themes) and "others, 24(6.7%)"; (3)Inadequate competency (where "low level of knowledge and staff negligence", 57(16.0%) and, "irregular staffs training", 21(5.9%) were the most dominated sub-themes; (4)Behavioral factors consisting of "resistance to change", 4(1.1%) and, "delay in reporting", 3(0.8%). Some of the extracts from the responses by the participants regarding challenges are provided below;

*"Not enough staff in the health information department and also inadequate computers" – Stated by a Medical Records Officer.*

*"…less knowledge on data quality concept in some profession especially nurses" – Stated by a Doctor.*

*"Inadequate training on data quality among some profession, e.g., nurses, doctors and lab technicians." – Stated by a Medical Records Officer*

*"There is poor internet connectivity to save data to cloud for storage and unstandardized data collection forms" – Stated by a Physician Assistant.*

*"Poor adherence to data collection and reporting guidelines, and resistance to change and cultural barriers" – Stated by a Laboratory Technician*

Underscoring the kind of impediments to ensuring data quality in the various healthcare facilities, the series of quoted statements reflects some of the salient responses captured from the participants during the study. Various types of professionals admitted to the existence of challenges, making optimal data-quality a far reached reality in their facilities.

In addressing the challenges impeding optimal data quality in healthcare facilities, six interconnected strategies were proposed by the HCP which were categorized under three main themes shown in Table 6: 1) Resources provision (more staff recruitment, 26.6%; enhancing EHR systems, 13.8% and improving internet and power supply, 5.2%); 2)

**Table 5. Perceived challenges hampering efforts to ensure data quality for information governance in healthcare facilities.**

| Category (Themes) | Code (Sub-Theme) | % Codes |
|---|---|---|
| *Poor data collection methods* | Data capturing errors and inconsistency | 22.40 |
| | Illegible handwriting | 1.70 |
| | Lack of standardization | 0.60 |
| *Inadequate Resources* | Less number of staffs | 24.40 |
| | Limited computer systems | 9.50 |
| | Poor Internet connectivity | 4.50 |
| | Others | 6.70 |
| *Inadequate Competency* | Low level of knowledge and staff negligence | 16.00 |
| | Irregular staff training | 5.90 |
| | Poor monitoring and supervision | 2.00 |
| *Behavioral Factors* | Resistance to change | 1.10 |
| | Delay in reporting | 0.80 |

**Source:** *Author's Analysis of Survey of Healthcare Professionals, 2024*

**Table 6. Strategies and best practices deployed to address challenges pertaining to ensuring data quality for information governance.**

| Category (Theme) | Code (Sub-Theme) | % Codes |
|---|---|---|
| *Resources Provision* | More staff recruitment | 26.60 |
| | Enhancing EHR systems | 13.80 |
| | Improving internet and power supply | 5.20 |
| *Competency Enhancement* | Education and training | 26.90 |
| *Effective monitoring and evaluation* | Regular data audit and supervision | 25.20 |
| | Enhancing information governance structure | 2.40 |

**Source:** *Author's Analysis of Survey of Healthcare Professionals, 2024*

Competency enhancement (education and training, 26.9%); 3) Effective monitoring and evaluation (regular data audit and supervision, 25.2% and enhancing information governance structures, 2.4%). Samples of the proposed strategies and best practices suggested by the participants are quoted below:

"*Adopting electronic devices in most wards to improve data collection*", – Stated by a Medical Records Officer

"*Organizing training and workshop for health information officers and other healthcare professionals*", – Stated by a Nurse

"*Efforts to provide necessary resources and provide more staffs in various departments*" – Stated by a Doctor.

"*Educating some profession on data quality and its importance as well as addressing internet and power supply issues*" – Stated by a Midwife

"*Regularly check all patient data to avoid inconsistency and make sure the patient data is accurately captured*" – Stated by an IT Officer.

Epitomizing strategies and bold solutions suggested by healthcare professionals, the quoted statement above are responses from the study participants. Reflecting from the quoted statement, professionals acknowledged major strategies and best practices towards the attainment of optimal data quality in our healthcare facility categorized into themes on Table 6.

### 3.1.4 Compliances and regulatory requirements related to health information governance

Of the 432 healthcare professionals involved in the study, 282(65.28%) attested to be familiar with key compliances and regulatory requirements related to health information governance. In addition to this, 133(30.79%) indicated being partially aware of these compliance and regulatory requirements while 17(3.94) admitted having no idea about these compliances and regulatory requirements.

Upon further interrogations, the HCP, on a multiple selection scale, identified the specific regulatory requirements or standards that when adhered to in the facility would enhance the governance of health information as indicated in Table 7. HIPAA, GDPR, FDA and HITECH Act were the regulatory requirements that were mostly advocated for with 361(33.7%), 228(21.3%), 185(17.3%) and 154(14.4%) proportions, respectively. Staying updated on these regulatory requirements, the mostly used methods in the healthcare facilities were the organization of regular training programs, 313(42.2%); internal compliance audits 179(24.1%) and, reading of newsletters or industry publications, 102(13.75%). In cases when there are data breaches, 82.6% of the HCP attested that there is an existing procedure for managing and reporting such suitable, while 3.2% admitted to be not sure of the existence of such procedure.

On the frequency at which compliance audits are conducted to the adherence to health information governance regulation, the healthcare professionals during the survey indicated that such compliance audits are conducted on a monthly

**Table 7. Various regulatory standards advocated by healthcare professionals for shaping the governance of health information in health facilities.**

| Regulatory Standard | Frequency (n = 428) | Percent |
|---|---|---|
| HIPAA | 361 | 33.68 |
| GDPR | 228 | 21.27 |
| HITECH Act | 154 | 14.37 |
| 21st Century Cures Act | 42 | 3.92 |
| FDA | 185 | 17.26 |
| ISO27001 | 102 | 9.51 |

**Source:** *Author's Analysis of Survey of Healthcare Professionals, 2024.*

basis, 303(70.1%). However, 37(8.6%), 23(5.3%) and 21(4.9%) also claimed such audits are sometimes conducted quarterly, weekly and annually respectively [Table 7].

### 3.1.5 Ethical considerations and privacy concerns associated with health data handling

The results of the study indicated that an IT Department and Ethics committee of the healthcare facilities are mostly responsible for overseeing and ensuring compliance with ethical guidelines and privacy regulations related to patient data handling with the following frequency and percentages 160(37.04%) and 159(36.81%). However, it was also believed that the efforts of these units in ensuring compliance with regulatory requirements are also rallied behind by the Management, 68(15.74%) and Privacy officers,45(10.42%).

Getting familiarized with ethical considerations and privacy associated with handling the sensitive data of patients as a healthcare professional is very crucial for effective information governance. In Table 8, measuring with a 1–5 familiarity index, the average level of familiarity with ethics and matters of privacy among the healthcare professionals during the survey was 3.13[3.02–3.25] with a standard error of 0.059.

Under three main facets (storage, processing & transmission), the measures deployed to ensure optimal privacy and confidentiality were probed into during the study. As displayed in Fig 2, during the storage of patient data five main different measures are deployed by healthcare professionals to ensure privacy and confidentiality of which the top three were authentication methods (use of passwords, biometric locks and pin codes), representing 58.9% of the total; physically securing the storage area, 12.5% and, using role-based access to stored patient data also representing 11.9%. On the account of processing the patient data, privacy and confidentiality are ensured by using secured computers and software for the processing, 71.2%; mounting effective supervision during processing, 16.2% and, also by making sure that data

**Table 8. Healthcare professionals level of familiarity with the ethical considerations and privacy concerns pertaining to handling health data and its dependence on professional type and years of experience.**

| Variable | Descriptive Statistics | | | |
|---|---|---|---|---|
| | *Scale* | *Mean* | *Std. Error* | *95% CI* |
| Familiarity with ethical considerations and privacy concerns associated with handling patient data | 1–5 | 3.13 | 0.059 | [3.02 – 3.25] |
| Demographic characteristics | **Test of Independence** | | | |
| | *Test Statistic ($\chi$2)* | *df* | *p-value* | *Cramer's V* |
| Profession type | 43.427 | 9 | 0.001** | 0.317 |
| Years of experience | 6.163 | 4 | 0.187 | 0.120 |

**Source:** *Author's Analysis Significant at P<0.001*

 

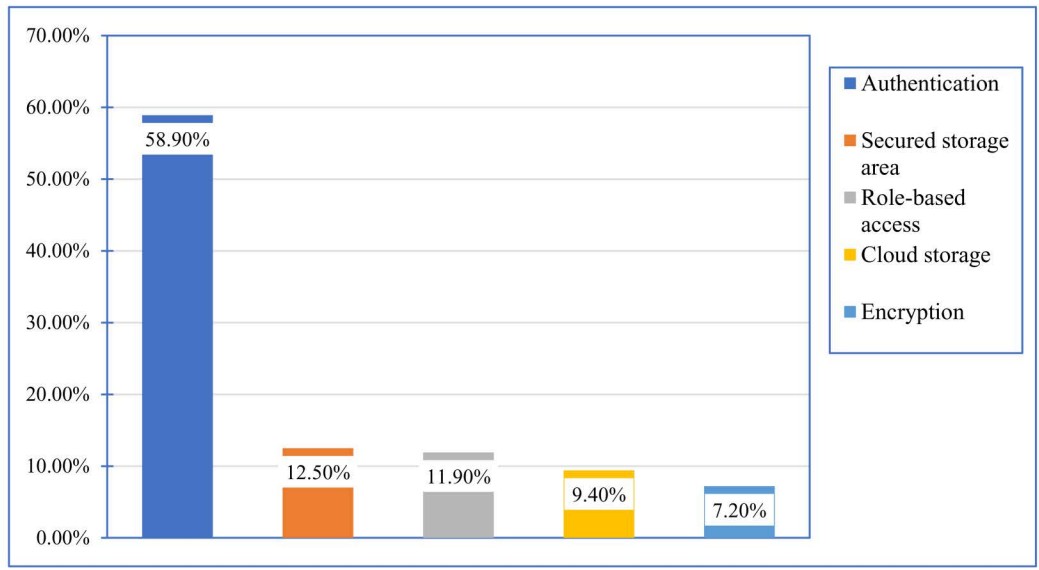

*Data Storage: Measures deployed to ensure privacy and confidentiality when storing patient data, where "authentication", "securing storage area" and 'role-based access" were the top three (3) measures used.*

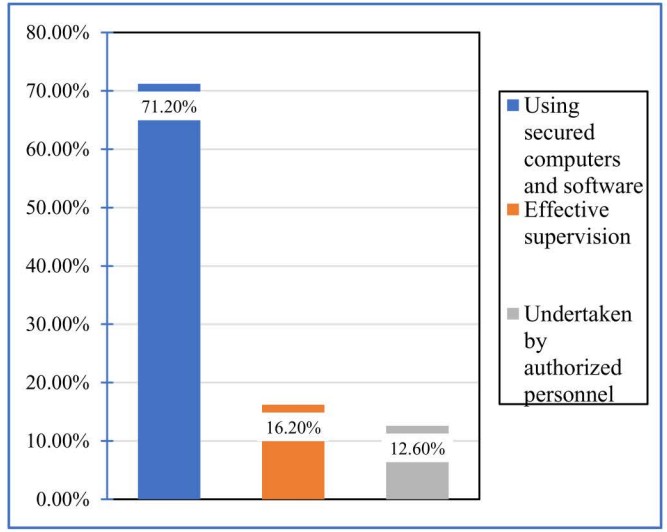

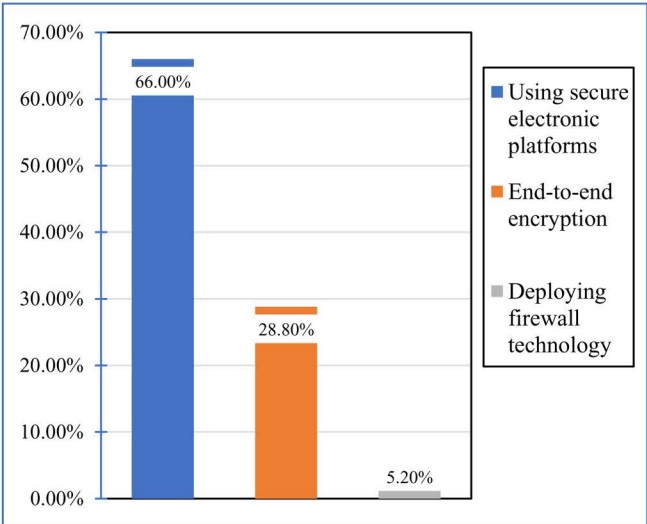

*Data processing: "Using secured computers & software" was the mostly used measure to ensure privacy and confidentiality during data processing*

*Data transmission: "Using secured electronic platforms" was the mostly used measure to ensure privacy and confidentiality during data transmission*

**Fig 2. Measures deployed to ensure the preservation of privacy and confidentiality when storing, processing and transmitting patient data as efforts towards good information governance.**

processing I executed by qualified personnel, 12.6%. During data transmission of patients' data within the facility or with a third-party privacy and confidentiality are ensured mainly by using secured electronic platforms, 66.0%. Also, end-to-end encryption is deployed in addition to firewall technologies to ensure that the transmitted data doesn't fall into an external

network with percentages of 28.8% and 5.2% respectively. In addition to these, HCP confirmed that explicit consent is sought from patients or data subjects before using data for other purposes rather than care delivery, 315(73.3%). However, 29(6.7%) were found to be unaware of sure informed consent is sought.

## 3.2 Discussion

### 3.2.1 Introduction and socio-demographic characteristics

The study probed into how effective data stewardship for health information governance is established and implemented in the arena of health. In conformity with the motive of the study, we also explored the best practices and strategies for managing health information, knowledge and adherence to compliance and regulatory requirements and also the level of understanding and practices related to privacy and confidentiality when handling patient data.

From a total of 432 healthcare professionals dominated by males, we garnered the requisite data geared towards answering the objective of the study. The recruitment encompassed nine (9) different professional types with nurses, health information officers and medical records officers being the top three dominant professional types within the survey. More than half of the recruited professionals were between the ages of 20–39 years and with approximately 5 years of average working experience (Table 1).

### 3.2.2 Effectiveness of the establishment and implementation of data stewardship for health information governance

Data stewardship in health information governance is a critical aspect of managing and safeguarding sensitive health data through proper management, oversight and accountability of data assets, and hence cannot be compromised [15–17]. With a clear understanding of data stewardship, the establishment of trust, adherence to regulatory requirements and maintaining data quality would be enhanced. Effective data stewardship practices not only ensure the accuracy and completeness of health data but also facilitate data sharing, interoperability, and decision-making processes within healthcare organizations [18–20]. The findings of this study imply that the majority of healthcare professionals have a moderate understanding of data stewardship and its related subjects (2.87/4.0) Unlike the findings of [21] & that of the findings of [22] in a study conducted in East Africa, where healthcare professionals showed a better understanding of data stewardship, this study's report suggests that only a few of the professionals have a clear understanding of subjects pertaining to data stewardship prerequisite for information governance. This echoes the findings of a study conducted in Keyan hospital which underscored major gaps in conjunction with the practice of advanced data stewardship [23]. However, the majority of the healthcare professionals during the study attested to the existence of clear policies and procedures guiding data stewardship, indicating a solid foundation for health information governance, as acknowledged by various scholars [24–26]. However, many of the healthcare lack a clear understanding of the said documentation, conflicting with the findings of [27]. The presence of a small fraction of respondents who believe no policies or procedures exist is particularly troubling, diverging from the findings of [28] where professionals had great levels of awareness. This could indicate either a complete lack of awareness or a perception that existing policies are ineffective or unenforced. For effective information governance in health, data stewardship should be a pivotal subject of interest [29].

Poor understanding of data stewardship according to our study, can be linked to the low rate of training sessions for professionals on data stewardship as unveiled in the study (1.86/4.0). In conformity with some obligations of healthcare professionals as posited by [30–32] in enhancing information governance, regular training enhances knowledge of effective data collection methods, storage and management of sensitive patient data. Findings from this study indicate that the rate at which training sessions are mounted for healthcare professionals is very low which consequently limits their understanding of such pertinent subject, antagonistic to the findings of [33] and [34] which indicated that regular training section on data management exceeds average. This unfortunate incidence of training deficit can be associated with the fact that, management and leadership of healthcare organizations trivialize data stewardship and data quality, hence reluctant to

invest into training and workshops within this arena. This mirrors [26] findings on resource constraints in African health-care settings, underscoring the need for sufficient allocations. Consequently, the quality of clinical decision-making, management level decision-making and the general quality of care would be adversely affected because these process hinge on quality data. In furtherance, our study revealed that the level of understanding and awareness of data stewardship practices is dependent on the professional type of the health worker. This study implies that the type of profession of a health worker has a very significant influence on their level of awareness and understanding of data stewardship as a subject of interest, affirming hypothesis 1 and failing to accept hypothesis 2. This also goes on to affirm the role-based understanding and awareness differences postulated by [35]. Similarly, our findings on this tangent mirrors other prior studies which highlighted the variation in knowledge on data stewardship been linked to professional type or role [36–38]. This may be due to the fact that some professional types such as health information officers and medical records officers might have specialized in such areas and hence, would be of good standing than those who did not, in matters relating to data stewardship. In the sense that data is generated and used by almost all healthcare professionals, it would be a step in the right direction when the level of awareness and understanding is prioritized and enhanced through regular organization of training, as postulated by [5]. Tailoring educational programs to meet the specific needs of different professions can help bridge these gaps and promote a more uniform understanding of health information governance across the organization.

### 3.2.3 Strategies and best practices for managing data quality for information governance initiatives

As posited in a scholarly article by [39], data quality is elucidated as "the extent to which data satisfies the prerequisites of its designated utility," The concept of data quality pertains to the precision, comprehensiveness, uniformity, and dependability of data contained in a dataset or database. Another exploration by [34] delineates data quality as "the degree to which data is suitable for its intended functions concerning accuracy, timeliness, completeness, uniformity, and pertinence." This comprehensive explanation accentuates the various facets of data quality that must be taken into account to guarantee its efficacy. It is of great importance that healthcare professionals become more familiar with the concepts of data quality and how significant it is for proper information governance [40,41]. Referencing the findings of this study, healthcare professionals' familiarity with the metrics of data quality was moderate and ought to be enhanced. The average level of familiarity with the concepts of data quality, which cannot be overlooked when craving for effective information governance as put forward by [42] was not encouraging among healthcare professionals during the survey. This result aligns with the findings of many other prior studies where familiarity and understanding of data quality were also identified to be challenged [43,44]. However, these results align with that of [45] who suggested that a basic understanding of data quality is essential for information governance. In the absence of robust data quality protocols, organizations run the risk of jeopardizing the credibility of their information, impeding operational efficiency, and eroding stakeholder confidence in the data employed for critical decision-making. As a ripple effect, the understanding of data quality metrics, leading suboptimal data quality can result huge financial lost, reduced productivity, misguided strategies and decisions as well as eroding stakeholders trust [46–48].

In light of this, our study's results concord with the idea of investing in data quality initiatives as put forward by [49]. In furtherance, a statistical test indicated that the level of familiarity with the concept of data quality is significantly influenced by the type of profession ($\chi2 = 18.364$, $p = 0.031$, Cramer's $V = 0.206$). Thus, there is an existence of a moderately significant relationship between profession and concept of data quality, leading to the affirmation of H3. Thus, understanding and getting familiarized with the concept of data quality was also seen to be dependent on the type of profession. Professionals who belong to departments which are directly responsible for data management are much more likely to be of greater understanding. This is an indication that at least conscious efforts are being made towards ensuring effective information governance in departments/units that are mainly responsible for data management, confirming the findings of [50].

As part of the conscious efforts geared towards augmenting the quality of data generated in the landscape of health, regularly conducting data quality assessments is very important. Mostly monthly, data quality assessments and audits are

conducted in various healthcare, as indicated by the results of this study. As acknowledged by [51], these monthly assessments allow organizations to maintain a consistent pulse on their data quality, enabling them to identify and address issues promptly, indicating a great approach to information governance. Aside, from the monthly data quality assessments, various departments and organizations as a whole might organize assessments on a weekly, bi-weekly, quarterly or annual basis depending on their peculiar protocols. Currently, our study identified Data Quality Assessments and Audits as a primary method for evaluating data quality. This approach aligns with the continuous monitoring and assessment strategies emphasized by [51]. Regular assessments allow healthcare organizations to systematically examine their data assets, identifying issues such as inaccuracies, inconsistencies, or incomplete information. Data Validation emerged as another significant method used by healthcare professionals. This aligns with the data quality rules and validation techniques discussed by [52]. Data validation processes help verify that information meets specific criteria or standards before it is used or entered into a system. This method is particularly effective in catching errors at the point of entry, thereby maintaining the overall quality of the data in the organization's systems. Electronically, the use of health information systems such as District Health Information Management systems (DHIMS) and Lightwave Health Information Management System (LHIMS) was also noted as a method for measuring and assessing the quality of data during capturing with the systems. These systems have a knowledge-based fed with predefined validation rules that trigger an alert when inconsistencies are identified during data entry.

Hampering the attainment of quality data in facilities are factors such as data capturing errors, limited number of staff and computer systems, low level of knowledge due to irregular training, resistance to changes, etc. as identified in this study (Table 4). This calls for proper restructuring of data collection methods, provision of resources and intensifying training sessions for effective information governance, parallel to the assertions of [53]. Conforming with the ideology of [54], there is a need for enhancing and promoting data literacy through training and education and, also intensifying regular data audits, monitoring and supervision for an early identification of errors and rectification as acknowledged by [55].

### 3.2.4 Compliances and regulatory requirements related to health information governance

According to the reports of this survey, a considerably higher proportion of healthcare professionals had great levels of familiarity with compliance and regulatory requirements about information governance, which is unparalleled to the findings of [43]. This widespread awareness aligns with the increasing emphasis on compliance in healthcare settings, as noted by other scholars [22,56], who highlight the growing focus on regulatory requirements among healthcare professionals, driven by the need to protect patient privacy and improve healthcare delivery quality. This result is highly positive because, compliance with regulatory requirements, such as data protection laws, industry standards and other data regulatory frameworks, is a critical aspect of information governance, as non-compliance can result in legal and financial consequences [16,57]. However, it's worth noting that some portion of healthcare professionals only admitted to being partially familiar or not familiar with these requirements. This suggests that while overall awareness is high, there may still be gaps in knowledge that need to be addressed, resonating with the finding of a survey of Ghanaian public hospitals, where inconsistent knowledge of regulatory requirements was identified as a potential barrier to effective information governance [58].

Specifically, healthcare professionals during the survey mostly advocated for adherence to the Health Insurance Portability and Accountability Act (HIPAA) which seeks to provide significant standards for protecting sensitive patient data as highlighted by [59]. In addition to this, another regulatory standard advocated by healthcare professionals during the survey was the General Data Protection Regulation (GDPR). As put forward by [60–62] GDPR highlights the importance of adopting a risk-based approach to ensure data identifiability and processing proportionality by incorporating technological safeguards. Again, the Health Information Technology for Economic and Clinical Health (HITECH) Act, which advocates for the utilization and meaningful integration of electronic health records (EHRs) and other health technology solutions as recognized by [63–65] was also highly advocated by the healthcare professionals. The high

levels of advocacy for these compliance and regulatory requirements present a vivid impression of how significant the professionals deem these standards, and hence how their adherence can improve information governance in general. The results of this study portray that regular training programs, internally conducted compliance audits, and reading of newsletters and industrial publications are the major methods through which healthcare professionals get updated on compliance and regulatory requirements. In addition to these, the study results provide a clear indication of how robust health information governance is established. This is evident in the regular conduct of monthly compliance audits to ascertain the adherence to health information governance regulation as attested by healthcare recruited during the survey.

### 3.2.5 Ethical considerations and privacy concerns associated with health data handling

Ethical considerations are paramount in the contemporary healthcare domain, as the management of patient data gives rise to notable ethical and privacy issues. The widespread utilization of electronic health records and digital health technologies has facilitated the gathering and dissemination of patient data, albeit escalating the potential for unauthorized access and misuse [66]. Healthcare entities and providers are then tasked with delicately balancing the utilization of patient data to enhance care and safeguard patient confidentiality [66]. In order not to compromise the privacy and confidentiality of sensitive patient data, healthcare professionals ought to be well-vested in ethical considerations and privacy concerns associated with handling patient data. The study's findings suggest that the majority of healthcare professionals are quite familiar with these pertinent concerns, however, the overall average on a 1 to 5 familiarity index was seen as moderate. Although, this implies a growing recognition of the importance of ethical consideration in data handling in healthcare, however, there is a need to invest efforts in enhancing the knowledge level of healthcare professionals. A statistical test at a 5% significance level indicated that the type of profession of health workers has a significant influence or association on their level of familiarity with ethical considerations and privacy concerns associated with handling patient data ($\chi2=43.427$, $p<0.001$, Cramer's V=0.317). In simple terms, this suggests that there is a significantly moderate relationship between profession, ethical considerations and privacy. As an implication, we accept H5 and fail to accept H6. As recounted in prior results session, profession type has a great influence on how familiar a healthcare professional would be with patient data privacy and confidentiality. Similarly, since some professionals such as health information officers have the direct mandate of managing patients' data, such professionals are likely to have much understanding and hence more familiarized with subjects of privacy and confidentiality when handling patients' data. As [67] argue, all healthcare professionals must adhere to ethical guidelines and regulations to maintain trust and integrity in their practice, hence there is the need to improve the level of familiarity as well.

During the storage of patient health data, secured authentication (use of a password, biometric verification, pin codes, etc.), physically securing the storage room and employing role-based assessment were some of the measures deployed in ensuring privacy and confidentiality as identified in our study. These align with what was put forward by [68,69] where they acknowledge the use of various authentication methods as a secured means of storing patient data. The use of secured computers and software during data processing was the major means by which privacy and confidentiality were ensured. Coupled with effective supervision and the use of qualified personnel, data processing is void of breach of privacy and confidentiality, affirming the assertion of [56] which advocates that only competent professionals should be allowed to process data, ensuring data privacy and confidentiality. Predominately, deploying secured electronic platforms during data transmission is data privacy and confidentiality are preserved. These secured platforms such as DHIMS, LHIMS and secured email systems use end-to-end encryption and firewall technologies, preventing unauthorized networks from tapping into patient data when being transmitted as postulated by [70,71]. These findings indicate organizations are taking steps to protect patient data privacy and confidentiality. As part of the study, we discovered that informed consent was sought from data subjects when intended to use their data beyond direct healthcare delivery, re-emphasizing healthcare professionals' deep regard for privacy and confidentiality.

### 3.2.6 Limitations

Data was collected from three different healthcare facilities, however, the self-reported data and rating deployed in the study could create significant bias in the general impression and insights derived from the study. In future studies, triangulate data sourcing would be used, including self-reports, reports from administration and direct observation studies to mitigate such biases in the study. Again, the study area includes only three (3) regions in Ghana, limiting its impressions from being generalized for all healthcare organizations, although results of the study are evident in prior studies emanating from different jurisdictions. Going forward, efforts would be geared towards including diverse sampling, taking into consideration multiple regions and cultural contexts. Also, rigorous comparative analysis would be done to ascertain the differences and similarities between various studies from different regions around the globe. Furthermore, the cross-sectional design permits the reliance on data collected in a short period, which is subjected to changes over time. Hence, the study only warrants temporal associations among variables. Addressing this, a longitudinal study is needed to validate cross-sectional findings in subsequent studies. Also, more advanced statistical analysis such as structural equation modelling would be incorporated in future studies to explore much detailed directional relationships between variables.

## 4.1 Conclusion

The results of the study indicate that healthcare organizations have data governance policies established, however, the understanding of these policies and procedures was moderate among professionals, signifying urgent education on these policies. Poor investment in training on data stewardship significantly limits healthcare professionals' level of understanding. Management and leadership poor posture on data stewardship significantly leads to less attention on data stewardship, hence, low rate of training on these pertinent issues which significantly affects the quality of care delivery and decision-making processes in healthcare facilities. Addressing the unfavorably stance of management and leadership on data-related issues would be a step in the right direction, enabling sufficient investment into ensuring data quality. The level of awareness and understanding on matters of data stewardship among some profession types was generally low and should be enhanced; some categories of professionals have much understanding of data stewardship than others. This skewness in knowledge affects the quality of data stewardship, given that, almost all healthcare professionals handle data in one way or the other. These knowledge gap can be mitigated through regular data workshops targeted at professionals who lack much understanding on data stewardship and identified in this study.

Healthcare organizations have implemented best practices to ensure data quality, enabling identification and early correction of data inconsistencies. However, these efforts are challenged by limited technological advancement, shortage of workforce, data capturing errors, low competencies and infrastructure issues such unstable internet and power supply. The fragile nature of digital health technology infrastructure affects the quality of data attained. Understanding of data quality concepts was generally low and varied by profession type. Whiles there was great recognition of the importance of regulatory requirements in shaping health information governance, knowledge of ethical consideration when handling patient data varied among professionals. Despite these gaps, calculated efforts are geared towards ensuring data privacy and confidentiality during all stages of data handling reflecting a positive posture for health information governance. The great variation in terms of level of awareness, understanding and knowledge existing among different professional types on these crucial subjects of health information governance warrants the need for improvement to strengthen the governance of health information in healthcare organizations.

On a global health settings level, the findings of this study underscore the urgent need for conscious efforts to enhance data stewardship in various healthcare settings. There is the need for a global standardized training and education on data stewardship to improve the understanding of professionals. International level organizations such as World Health Organization (WHO) and Healthcare Information and Management Systems Society (HIMSS) could develop more rounded guidelines, ensuring that healthcare professionals are consistently and effectively educated on data stewardship.

The global health systems must prioritize workforce capacity building targeted at ensuring high-quality data management in healthcare. This will reduce the huge knowledge gap among different healthcare professionals to ensure that all healthcare professionals, irrespective of their roles, handle data adequately. Furthermore, there is the need for internationally collaborated efforts to strengthen digital health infrastructure, addressing challenges such as unstable internet and power supply in low-resource areas. Moreover, the findings of the study necessitate the need for an internationally harmonized information governance systems ensuring interoperability and secure information exchange between countries. Addressing these concerns on the global level would ultimately revamp patient care and health outcomes worldwide.

## 4.2 Recommendations

To enhance the effectiveness of information governance, it is recommended based on this study that:

Healthcare organization should prioritize training sessions on how to ensure data quality and proper data stewardship should be organized regularly to bridge the knowledge gap among some professionals. Special emphasize should be placed on how to ensure confidentiality, privacy and avert data breaches in this current computer dominated healthcare, revamping the governance of data. The training sessions should be professionally centered, giving emphasis to each professional's type to easily identify their weaknesses and improve on them. Regular training can also be used as a means for updating healthcare professionals on regulatory requirements.

Management and leadership of healthcare organizations should develop clear and concise data governance policies that outline roles, responsibilities, and procedures for data collection, storage, access, and sharing. Ensuring the compliances of these policies, policy-makers should deploy dedicated committee that would enforce that healthcare professionals toe the line pertaining to data governance. Also, by deploying effective communication channels, hospital management should make data governance policies and procedures available and comprehensive enough to all healthcare professionals.

Policymakers should establish a schedule for reviewing and updating data governance policies to reflect changes in regulations, technology, and organizational practices from the national level, transcending to the organizational level. Again, routine audits of data management practices should be implemented to identify weaknesses and areas for improvement using the findings of various studies as guide to inform training and policy updates, ensuring the organization continuously evolves in its data governance efforts.

Researchers should focus on delving into studies to unveil the elapses in knowledge among healthcare professionals on data stewardship and governance by employing longitudinal and experimental designs involving multiple regions and jurisdiction to warrant deeply rooted conclusions. As a result, this would help streamline training sessions to a targeted profession type, fast-tracking data governance improvement.

## Acknowledgments

We thank the management and staff of the participatory hospitals for allowing us to undertake this study and are grateful to all individuals who contributed to its success.

## Author contributions

**Conceptualization:** Richard Okyere Boadu, Kwame Adu Okyere Boadu.

**Data curation:** Richard Okyere Boadu, Victor Wireko Adu.

**Formal analysis:** Richard Okyere Boadu, Kwame Adu Okyere Boadu, Victor Wireko Adu, Baishatu Ibrahim, Prince Akey, Amponsah Amishadas Mensah.

**Investigation:** Richard Okyere Boadu, Kwame Adu Okyere Boadu, Baishatu Ibrahim, Prince Akey, Amponsah Amishadas Mensah.

**Methodology:** Richard Okyere Boadu, Baishatu Ibrahim, Prince Akey, Amponsah Amishadas Mensah.

**Project administration:** Richard Okyere Boadu.

**Resources:** Richard Okyere Boadu.

**Software:** Richard Okyere Boadu, Victor Wireko Adu.

**Supervision:** Richard Okyere Boadu.

**Validation:** Richard Okyere Boadu.

**Visualization:** Richard Okyere Boadu, Victor Wireko Adu.

**Writing – original draft:** Richard Okyere Boadu, Victor Wireko Adu.

**Writing – review & editing:** Richard Okyere Boadu, Kwame Adu Okyere Boadu, Baishatu Ibrahim, Prince Akey, Amponsah Amishadas Mensah, Nathan Kumasenu Mensah, Godwin Adzakpah.

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
