## [Decision Letter · Decision Letter 0]

PONE-D-24-51872EXAMINE FRAMEWORKS POLICIES, AND STRATEGIES FOR EFFECTIVE INFORMATION GOVERNANCE IN HEALTHCARE ORGANIZATIONSPLOS ONE

Dear Dr. Okyere Boadu,

Thank you for submitting your manuscript to PLOS ONE. After careful consideration, we feel that it has merit but does not fully meet PLOS ONE’s publication criteria as it currently stands. Therefore, we invite you to submit a revised version of the manuscript that addresses the points raised during the review process.

**ACADEMIC EDITOR: **

The manuscript is generally well-written, and the topic is significant as it highlights the importance of examining framework policies to guide data stewardship and enhance overall information governance. However, there are several major concerns that require the authors' close attention to improve the overall quality of the manuscript.

The abstract is overly detailed, with some areas feeling cluttered. While it provides a lot of information, key results and implications are not highlighted effectively. Simplify language for better readability.

While the manuscript is well-focused in Ghana, the global significance of the findings is underexplored. This limits its broader relevance to international readers. Add a comparison of the findings with global standards or practices in information governance (e.g., from WHO, developed nations like the UK/US).

The methodology:

Include a detailed explanation of how the sample was randomized (e.g., stratified random sampling?).Justify the choice of a semi-structured questionnaire by linking it to study objectives (e.g., balance of quantitative and qualitative insights).Clarify whether written informed consent was obtained and if anonymity was guaranteed.Specify how sensitive health information was handled to ensure privacy during data collection and analysis.

Results section:

Some statistical tests (e.g., chi-square tests) are reported, but effect sizes and confidence intervals are not always included. This might reduce the perceived rigor. Consistently report effect sizes and confidence intervals for all statistical tests.Clearly interpret the statistical results for non-technical readers (e.g., “This suggests a moderate relationship between profession and understanding of data stewardship”).Provide richer qualitative data to contextualize numerical findings. For example, quote participants describing barriers to data quality (e.g., “One participant noted, ‘Inconsistent data formats make it impossible to standardize records.’”).Simplify tables (Table 5 and Table 7) by summarizing key takeaways rather than overloading with numbers.Ensure all charts have detailed captions and legends for clarity. For instance, specify what “Measures for Privacy” include in Figure 1.

Discussion section:

The results indicate low levels of training on data stewardship (1.86/4.0). However, the manuscript doesn’t deeply explore why this training deficit exists or its implications.Explicitly address limitations such as:Potential biases in data collection (e.g., self-reported data).Limited generalizability to other regions or countries.Challenges in data accuracy due to reliance on cross-sectional design.
Propose specific actions to mitigate these issues in future research.While the manuscript mentions training and policy updates, specific recommendations for healthcare organizations, policymakers, or researchers are vague.

Conclusion section

The conclusion is understated and doesn’t sufficiently tie together the findings and implications.

We look forward to receiving your revised manuscript.

Kind regards,

Mohd Ismail Ibrahim, MCom.Med

Academic Editor

PLOS ONE

2. In the ethics statement in the Methods, you have specified that verbal consent was obtained. Please provide additional details regarding how this consent was documented and witnessed, and state whether this was approved by the IRB

3. In the online submission form you indicate that your data is not available for proprietary reasons and have provided a contact point for accessing this data. Please note that your current contact point is a co-author on this manuscript. According to our Data Policy, the contact point must not be an author on the manuscript and must be an institutional contact, ideally not an individual. Please revise your data statement to a non-author institutional point of contact, such as a data access or ethics committee, and send this to us via return email. Please also include contact information for the third party organization, and please include the full citation of where the data can be found.

4. Please upload a copy of Figure 3.1, to which you refer in your text on page 12. If the figure is no longer to be included as part of the submission please remove all reference to it within the text.

Reviewers' comments:

Reviewer's Responses to Questions

**Comments to the Author**

1. Is the manuscript technically sound, and do the data support the conclusions?

Reviewer #1: Yes

2. Has the statistical analysis been performed appropriately and rigorously? 

Reviewer #1: Yes

3. Have the authors made all data underlying the findings in their manuscript fully available?

Reviewer #1: Yes

4. Is the manuscript presented in an intelligible fashion and written in standard English?

Reviewer #1: Yes

5. Review Comments to the Author

Reviewer #1: Feedback on Manuscript Quality Across Sections

1. Introduction

Strengths:

The introduction provides a clear rationale for the study, emphasizing the importance of health information governance (IG) in healthcare.

It cites relevant literature to contextualize the research problem and the need for effective IG frameworks.

Areas for Improvement:

Clarity and Focus: While the introduction is informative, some paragraphs are overly dense and repetitive. Simplify to enhance readability.

Gap Identification: Explicitly state how the study addresses gaps in the existing literature on IG frameworks in healthcare.

Specificity: Include recent statistics or case studies to strengthen the relevance of the research.

2. Methodology

Strengths:

The study design is appropriate (cross-sectional) for the objectives and includes a well-documented sample of healthcare professionals.

Analytical methods for qualitative and quantitative data are robust, employing tools like Stata and Qualitative Data Analysis Miner.

Areas for Improvement:

Data Collection Tool Description: Provide more detail on the semi-structured questionnaire’s development, validity testing, and pre-testing, if applicable.

Sampling Technique: While the random sampling approach is described, include a justification for the sample size calculation and the representativeness of the population.

Ethical Considerations: Clarify the consent process (e.g., written or verbal) and provide more details on how anonymity and confidentiality were ensured.

3. Results

Strengths:

Results are well-organized, with clear subheadings for each research question.

Tables and charts effectively summarize key findings, making the data accessible.

Areas for Improvement:

Interpretation: Provide more in-depth interpretations of key findings within the results section itself to help readers understand their significance.

Details on Subgroups: Expand on subgroup analyses, such as differences among professional roles, to add depth to the findings.

Figure Titles: Use more descriptive titles and captions to ensure figures and tables are understandable without referring back to the text.

4. Discussion

Strengths:

The discussion aligns findings with existing literature, reinforcing the study's contributions.

It identifies practical implications for improving IG in healthcare organizations.

Areas for Improvement:

Novelty: Highlight the unique contributions of this study to the IG literature more explicitly.

Limitations: Expand on study limitations, such as potential biases in self-reported data and the cross-sectional design.

Actionable Recommendations: Suggest more specific strategies or interventions for enhancing IG in healthcare, tailored to low-resource settings.

5. Conclusion

Strengths:

The conclusion effectively summarizes the study's key findings and their implications for healthcare IG.

Recommendations are practical and align well with the study objectives.

Areas for Improvement:

Conciseness: Streamline the conclusion to focus on the most critical findings and recommendations.

Forward-Looking Statements: Include suggestions for future research to address remaining gaps.

General Feedback

Writing Quality

Strengths:

The manuscript is generally well-written, with a formal academic tone.

Technical terminology is used appropriately, reflecting the study's complexity.

Improvements:

Address grammatical errors (e.g., "less number of staffs" should be "fewer staff members") and typographical mistakes.

Reduce verbosity in some sections, especially the introduction and discussion.

Specific Areas for Improvement

Abstract: Simplify and condense the abstract for clarity, ensuring all critical information (e.g., major findings, implications) is included within the word limit.

Consistency: Ensure consistent formatting and terminology throughout (e.g., “data governance” vs. “information governance”).

6. PLOS authors have the option to publish the peer review history of their article (what does this mean? ). If published, this will include your full peer review and any attached files.

**Do you want your identity to be public for this peer review?** For information about this choice, including consent withdrawal, please see our Privacy Policy .

Reviewer #1: No

---

## [Author Response · Author response to Decision Letter 1]

20 Jan 2025

15 Jan. 2025

Editor-in-Chief

PLOS ONE

Dear Sir/Madam,

Re: Responses to Reviewers and Editor’s Comments

On behalf of my colleagues, I am submitting responses to Reviewers and Editor’s comments raised in our article “Examine frameworks policies, and strategies for effective information governance in healthcare organizations”.

The following areas of concern raised by the reviewers and Editor with responses (highlighted) as detailed below:

Abstract

1. The abstract is overly detailed, with some areas feeling cluttered. While it provides a lot of information, key results and implications are not highlighted effectively. Simplify language for better readability.

Thank you for your feedback. The results aspect of the abstract has been revised to read, “The level of understanding of data stewardship (2.87/4.0), familiarity with concepts of data quality (2.73/5.0) and ethical considerations and privacy (3.13/5.0) when dealing with patient health data were found to be varied among the professionals, overall rated as moderate and affirming the need for improvement. Level of knowledge and understanding on these pertinent areas were highly influenced by type of profession (p<0.001; p=0.031 and, p<0.001) respectively and been indicative that some type of health workers is much concerned about data stewardship and health information governance than others. Various methods and efforts were in place to ensure quality of data, however, hampering these attempt included factors such as poor data collection methods, lack of resources, inadequate competency and some behavioural factors…”[see lines 29-55]

Methodology

2. Include a detailed explanation of how the sample was randomized (e.g., stratified random sampling?).

Thank you for your feedback. The detailed explanation to the sampling procedure has been included to the methodology to read, “…From a stratification based on the category of profession, a simple random sampling technique was deployed in drawing the sample from the targeted population of healthcare professionals in the various health facilities. A proportion of HCP were selected from every stratum through randomization of staff ID, giving equal opportunity to partaking in the study”.[see lines 185-189]

3. Justify the choice of a semi-structured questionnaire by linking it to study objectives (e.g., balance of quantitative and qualitative insights).

Thank you for your feedback. Justification for employing semi-structured questionnaire have been included and also linked to the objectives of the study, which reads, “Closed-ended questions represented quantitative part of the study whiles opened-ended questions also intended for qualitative aspect,…Specifically, opened-ended questions solicited for qualitative data on current methods used for assessing and measuring data quality, data quality challenges, and the strategies and best practices deployed to address the challenges (all under section/objective 3 of the study). In addition, qualitative data were garnered under section 5, where HCP professionals gave measures/strategies employed by their organisation during the storage, transmission, and processing of patient data.” [see lines 192-193 & 198-203]

4. Clarify whether written informed consent was obtained and if anonymity was guaranteed.

Thank you for your feedback. Clarification on the type of informed consent obtained and how optimal anonymity is ensured has been provided. [see lines 217-221]

5. Specify how sensitive health information was handled to ensure privacy during data collection and analysis.

Thank you for your feedback. Information on how data privacy was ensured during collection and analysis has been specified. [see lines 220-223]

Results Section

6. Some statistical tests (e.g., chi-square tests) are reported, but effect sizes and confidence intervals are not always included. This might reduce the perceived rigor. Consistently report effect sizes and confidence intervals for all statistical tests.

Thank you for your feedback. The results has been revised to include effect size and confidence intervals for all statistical tests. [ see table 3, 4 & 8]

7. Clearly interpret the statistical results for non-technical readers (e.g., “This suggests a moderate relationship between profession and understanding of data stewardship”).

Thank you for your feedback. Interpretations to the result of all three statistical test have been revised to make them clear. [see lines 271-273 , 298-304 & 451-453 for all each interpretation respectively]

8. Provide richer qualitative data to contextualize numerical findings. For example, quote participants describing barriers to data quality (e.g., “One participant noted, ‘Inconsistent data formats make it impossible to standardize records.’”).

Thank you for your feedback. Various extracts have been added to sections of the results that involved qualitative method. [see lines 321-334, 347-357 & 381-391 for the respective quality data]

9. Simplify tables (Table 5 and Table 7) by summarizing key takeaways rather than overloading with numbers.

Thank you for your feedback. The content of the tables have been summarized accordingly.[see tables 5 & 7]

10. Ensure all charts have detailed captions and legends for clarity. For instance, specify what “Measures for Privacy” include in Figure 1.

Thank you for your feedback. All the charts in figure 1 have been revised accordingly to include captions and legends. [see lines 493-496 & figure 1]

Discussion section

11. The results indicate low levels of training on data stewardship (1.86/4.0). However, the manuscript doesn’t deeply explore why this training deficit exists or its implications.

Thank you for your feedback. A section of the decision has been revised to explore the possible cause of this deficit as “…This unfortunate incidence of training deficit can be associated with the fact that, management and leadership of healthcare organizations trivialize data stewardship and data quality, hence reluctant to invest into training and workshops within this arena. Consequently, the quality of clinical decision-making, management level decision-making and the general quality of care would be adversely affected because these process hinge on quality data.” [see lines 551-555]

12. Explicitly address limitations such as:

a. Potential biases in data collection (e.g., self-reported data).

b. Limited generalizability to other regions or countries.

c. Challenges in data accuracy due to reliance on cross-sectional design.

Thank you for your feedback. A section has been included to address all the limitations of the study. [see lines 683-684, 686-689 & 692-693]

13. Propose specific actions to mitigate these issues in future research.

Thank you for your feedback. Specific mitigations strategies for future research have been added to address all the study limitations. [see lines 685-686, 689-691 & 694-696]

14. While the manuscript mentions training and policy updates, specific recommendations for healthcare organizations, policymakers, or researchers are vague.

Thank you for your feedback. The recommendations section has been revised to address these concerns. [see lines 727-858]

Conclusion section

15. The conclusion is understated and doesn’t sufficiently tie together the findings and implications.

Thank you for your feedback. The conclusion section has been revised to sufficiently capture the findings of the study. The revised conclusion reads as follows “The results of the study indicates that healthcare organizations have data governance policies established, however, the understanding of these policies and procedures was moderate. Poor investment in training on data stewardship significantly limits healthcare professionals’ level of understanding. Management and leadership poor posture on data stewardship significant leads to less attention on data stewardship, hence, low rate of training on these pertinent issues which significantly affects the quality of care delivery and decision-making processes in healthcare facilities. The level of awareness and understanding on matter of data stewardship among some profession types should be enhanced; some categories of professionals have much understanding of data stewardship than others. This skewness in knowledge affect the quality of data stewardship given that, almost all healthcare professionals handle data in one way or the other. Moreover, there are some best practices put in place to ensure data quality,…” [see lines 698-718]

---

## [Decision Letter · Decision Letter 1]

PONE-D-24-51872R1EXAMINE FRAMEWORKS POLICIES, AND STRATEGIES FOR EFFECTIVE INFORMATION GOVERNANCE IN HEALTHCARE ORGANIZATIONSPLOS ONE

Dear Dr. Okyere Boadu,

Thank you for submitting your manuscript to PLOS ONE. After careful consideration, we feel that it has merit but does not fully meet PLOS ONE’s publication criteria as it currently stands. Therefore, we invite you to submit a revised version of the manuscript that addresses the points raised during the review process.

**I have carefully reviewed your revised manuscript. While I acknowledge that significant improvements have been made in response to the reviewers' comments, there are still areas where minor adjustments could enhance the clarity and overall quality of the manuscript.**

**Minor issues:**

**There are minor grammatical errors and awkward phrasing in some sections, particularly in the discussion. A language editor could help improve clarity and flow.****While limitations are addressed, the discussion of the implications for global health settings could be expanded. For example, how applicable are these findings beyond Ghana?****Although qualitative data was included, more direct participant quotes could add depth and improve the contextualization of findings.****The conclusion could more strongly tie the findings to actionable policy recommendations.****Some sections repeat earlier information, particularly between the abstract and introduction.**

We look forward to receiving your revised manuscript.

Kind regards,

Mohd Ismail Ibrahim, MCom.Med

Academic Editor

PLOS ONE

**Journal Requirements:**

Reviewers' comments:

Reviewer's Responses to Questions

**Comments to the Author**

1. If the authors have adequately addressed your comments raised in a previous round of review and you feel that this manuscript is now acceptable for publication, you may indicate that here to bypass the “Comments to the Author” section, enter your conflict of interest statement in the “Confidential to Editor” section, and submit your "Accept" recommendation.

Reviewer #2: All comments have been addressed

2. Is the manuscript technically sound, and do the data support the conclusions?

Reviewer #2: Partly

3. Has the statistical analysis been performed appropriately and rigorously? 

Reviewer #2: Yes

4. Have the authors made all data underlying the findings in their manuscript fully available?

Reviewer #2: Yes

5. Is the manuscript presented in an intelligible fashion and written in standard English?

Reviewer #2: No

6. Review Comments to the Author

**Reviewer #2: ** 1. The revised paper shows that all the previous comments have been addressed. However, I don't think the manuscript is yet in its best to be published.

2. The manuscript is partly technically sound. The results do align with the mentioned objective of the study. However, I find the result section a little unclear. There's no need to provide any explanation for the results obtained (in the result section). The assumptions for the result can be later installed in the discussion section. The description for the tables needs to be more precise for easy understanding of the reader irrespective of their professions. The discussion part needs more supporting references with comparisons. The qualitative portion of the manuscript too many back-to-back quotations without much reasoning/explanation behind using that statement as a quotation. I would also recommend using "Stated by a Doctor/Nurse/Any other healthcare professional" used at the end of each quotation referring to who might have provided this statement. The qualitative part of the manuscript requires some detailing to strengthen the paper.

3. Yes, I believe the statistical analysis has been performed properly.

4. Yes, the authors have made the data underlying the findings in their manuscript fully available to the editor.

5. The manuscript is very poorly written in English. The author should put more effort in rectifying the jargons in the sentences and try to simplify them. The author needs to thoroughly re-read the entire manuscript and correct the silly punctuation mistakes. Some sentences are not making any sense due to usage of unnecessary complicated words. Such sentences should be re-written to make meaningful clear statements.

7. PLOS authors have the option to publish the peer review history of their article (what does this mean? ). If published, this will include your full peer review and any attached files.

**Do you want your identity to be public for this peer review?** For information about this choice, including consent withdrawal, please see our Privacy Policy .

Reviewer #2: No

---

## [Author Response · Author response to Decision Letter 2]

18 Mar 2025

13th March, 2025

Editor-in-Chief

PLOS ONE

Dear Sir/Madam,

Re: Responses to Reviewers' and Editor’s Comments

On behalf of my colleagues, I am submitting responses to the Reviewers and Editor’s comments raised in our article “EXAMINE FRAMEWORKS POLICIES, AND STRATEGIES FOR EFFECTIVE INFORMATION GOVERNANCE IN HEALTHCARE ORGANIZATIONS”.

The following areas of concern raised by the reviewers and Editor with responses (highlighted) as detailed below:

1. There are minor grammatical errors and awkward phrasing in some sections, particularly in the discussion. A language editor could help improve clarity and flow.

Thank you for the feedback. The grammatical errors have been rectified as recommended.

2. While limitations are addressed, the discussion of the implications for global health settings could be expanded. For example, how applicable are these findings beyond Ghana?

Thank you for the feedback. Detailed applications of the findings have been added to the conclusion section of the manuscript [see line 747 – 761].

3. Although qualitative data was included, more direct participant quotes could add depth and improve the contextualization of findings.

Thank you for the feedback. Additional direct participant quotes have been added to the various sections [see lines 299-300, 306-307, 315-17, 346-347& 387-388].

4. The conclusion could more strongly tie the findings to actionable policy recommendations.

Thank you for the feedback. The conclusion section has been revised accordingly [see lines 719- 765].

5. Some sections repeat earlier information, particularly between the abstract and introduction.

Thank you for the feedback. The abstract has been revised to do away with the repeated information [see lines 16 – 19].

6. The description for the tables needs to be more precise for easy understanding of the reader irrespective of their professions.

Thank you for the feedback. The description of the various tables have been revised [see lines 255-256, 269-271, 362-363, 407-408 & 439-441]

7. The qualitative portion of the manuscript too many back-to-back quotations without much reasoning/explanation behind using that statement as a quotation. I would also recommend using "Stated by a Doctor/Nurse/Any other healthcare professional" used at the end of each quotation referring to who might have provided this statement.

Thank you for the feedback. The recommended changes have been made to the quotations of the qualitative sections. [see lines 296-327, 340-358 & 375-393].

8. The discussion part needs more supporting references with comparisons.

Thank you for the feedback. The discussion section has been revised with additional references and comparisons [see lines 506-585 & 623-644, also see references section for their corresponding bibliographies].

9. However, I find the result section a little unclear. There's no need to provide any explanation for the results obtained (in the result section). The assumptions for the result can be later installed in the discussion section

Thank you for the feedback. For the purpose of understanding by a non-technical reader, the explanation and assumptions of the results are both highlighted in the results and discussion sections. This is done to enhance immediate understanding of the results by a reader [see lines 544-548, 582-585 & 666-67],

Thank you for your reconsideration and re-evaluation prior to consideration for publication. My colleagues and I appreciate your time and effort and look forward to hearing from you.

Sincerely,

Richard Okyere Boadu (PhD)

Department of Health Information Management

School of Allied Health Sciences

College of Health and Allied Health Sciences

University of Cape Coast

Cape Coast, Ghana

richard.boadu@ucc.edu.gh

---

## [Decision Letter · Decision Letter 2]

EXAMINE FRAMEWORKS POLICIES AND STRATEGIES FOR EFFECTIVE INFORMATION GOVERNANCE IN HEALTHCARE ORGANIZATIONS

PONE-D-24-51872R2

Dear Dr. Okyere Boadu,

We’re pleased to inform you that your manuscript has been judged scientifically suitable for publication and will be formally accepted for publication once it meets all outstanding technical requirements.

Kind regards,

Mohd Ismail Ibrahim, MCom.Med

Academic Editor

PLOS ONE

Additional Editor Comments (optional):

Reviewers' comments:

Reviewer's Responses to Questions

**Comments to the Author**

1. If the authors have adequately addressed your comments raised in a previous round of review and you feel that this manuscript is now acceptable for publication, you may indicate that here to bypass the “Comments to the Author” section, enter your conflict of interest statement in the “Confidential to Editor” section, and submit your "Accept" recommendation.

Reviewer #3: All comments have been addressed

Reviewer #4: All comments have been addressed

Reviewer #5: All comments have been addressed

2. Is the manuscript technically sound, and do the data support the conclusions?

Reviewer #3: Yes

Reviewer #4: Yes

Reviewer #5: Yes

3. Has the statistical analysis been performed appropriately and rigorously? 

Reviewer #3: Yes

Reviewer #4: Yes

Reviewer #5: Yes

4. Have the authors made all data underlying the findings in their manuscript fully available?

Reviewer #3: Yes

Reviewer #4: Yes

Reviewer #5: No

5. Is the manuscript presented in an intelligible fashion and written in standard English?

Reviewer #3: Yes

Reviewer #4: Yes

Reviewer #5: Yes

6. Review Comments to the Author

Reviewer #3: The document reads well to me. The concepts and their presentation in my view has been done well. Line 670 mentions "elapses" - I guess that should read as "lapses". Thank you!!!

Reviewer #4: All comments have been addressed by the Author accordingly. The manuscript may be proceeded for publication.

Reviewer #5: This is an interesting paper that explores an important area of information governance. It is clear that the paper has already gone through rounds of revisions and that the authors have fully engaged with the comments of past reviewers, substantively improving the paper itself.

That being said, there are a couple of areas that I believe could be further developed. Firstly, and in line with previous comments, there remains a number of small typos throughout the paper, along with some odd word choices. For instance, the title does not entirely read well, with "examining frameworks, policies, and strategies…" making more sense than the current version, which serves more as an instruction to "examine". There is also the coded sub-theme of "less number of staffs", which does not entirely scan. The apparent mis-spelling of "Kenyan" on line 503. And the seeming overuse of a thesaurus when comparing the results to prior work, such as "antagonistic to the findings of…" and "parallel to the assertions of…", etc.

Indeed (and secondly), when comparing the results to prior work, there is an extended period when the findings herein do not conform to extant literature, but no effort is made to explain why this may be the case.

Thirdly, throughout the paper, there is a mix between using scores and percentages, where converting all of these to percentages may make it easier to read. Similarly, with these scores, it is unclear why one scale is 1-5, when the next is 1-4.

Finally, line 528 suggests that "awareness of data stewardship practices is dependent on the professional type of health worker". However, while a link has clearly been established, I am not convinced the paper can justify the claim that awareness is *dependent* on worker type.

7. PLOS authors have the option to publish the peer review history of their article (what does this mean? ). If published, this will include your full peer review and any attached files.

**Do you want your identity to be public for this peer review?** For information about this choice, including consent withdrawal, please see our Privacy Policy .

Reviewer #3: No

Reviewer #4: **Yes: ** Dr. Jamil Afzal

Reviewer #5: No

---

## [Editor Report · Acceptance letter]

PONE-D-24-51872R2

PLOS ONE

Dear Dr. Okyere Boadu,

I'm pleased to inform you that your manuscript has been deemed suitable for publication in PLOS ONE. Congratulations! Your manuscript is now being handed over to our production team.

Kind regards,

on behalf of

Dr. Mohd Ismail Ibrahim

Academic Editor

PLOS ONE